

# Cascade effect of rock bridge failure in planar rock slides: explicit numerical modelling with a distinct element code

Adeline Delonca[1], Yann Gunzburger[2], Thierry Verdel[2]

[1] Departamento de Ingeniería Metalúrgica y de Materiales (DIMM), Universidad Técnica Federico Santa Maria, Campus San Joaquín, Santiago, Chile
[2] GeoRessources, UMR 7359, Université de Lorraine - CNRS, Ecole des Mines de Nancy, Campus ARTEM, BP14234 FR-54042 Nancy Cedex, France

*Correspondence to*: Adeline Delonca (adeline.delonca@usm.cl)

**Abstract.** Plane failure along inclined joints is a classical mechanism involved in rock slopes movements. It is known that the number, size and position of rock bridges along the potential failure plane are of main importance when assessing slope stability. However, the rock bridges failure phenomenology itself has not been comprehensively understood up to now. In this study, the propagation cascade effect of rock bridges failure leading to catastrophic block sliding is studied and the influence of rock bridges position in regard to the rockfall failure mode (shear or tension) is highlighted. Numerical modelling using the distinct element method (UDEC-ITASCA) is undertaken in order to assess the stability of a 10 m$^3$ rock block lying on an inclined joint with a dip angle of 40° or 80°. The progressive failure of rock bridges is simulated assuming a Mohr-Coulomb failure criterion and considering stress transfers from a failed bridge to the surrounding ones. Two phases of the failure process are described: (1) a stable propagation of the rock bridge failures along the joint and (2) an unstable propagation (cascade effect) of rock bridges failures until the block slides down. Additionally, the most critical position of rock bridges has been identified. It corresponds to the top of the rock block for a dip angle of 40° and to its bottom for an angle of 80°.

## 1 Introduction

Rockfall hazard is defined as "the probability of occurrence of a potentially damaging rockfall within a given area and in a given period of time" (Varnes 1984). The damaging phenomenon generally results from the failure of weakness planes and the fall of one or several rock blocks down to the target area (Corominas et al. 2005). On other words, the rockfall hazard can be defined as the failure probability multiplied by the probability of propagation. While different probabilistic methods exist to calculate the probability of propagation (Guzzetti et al. 2002; Jaboyedoff, Dudt, and Labiouse 2005; Bourrier et al. 2009; Levy et al. 2018), the failure probability is more complex to assess. Methods mainly based on expert judgment (Delonca, Verdel, and Gunzburger 2016) and empirical methods (Jaboyedoff, Dudt, and Labiouse 2005; Mazzoccola and Hudson 1996; Dussauge-Peisser et al. 2002) are mostly used to date, but they do not consider the failure mechanism leading to the triggering of an event. Statistical analysis (Chau et al. 2003; Coe et al. 2004; Delonca, Gunzburger, and Verdel 2014) can also be used to approach the temporality of the hazard, but presents the same restriction than the other methods. However, the understanding





of the failure process of weakness planes is a major issue for risk assessment as it is responsible for the generation of a rockfall and defines its time of occurrence.

The main parameter controlling the resistance of a rock joint, and therefore the failure mechanism, are the rock bridges ("International Society for Rock Mechanics Commission on Standardization of Laboratory and Field Tests: Suggested

Methods for the Quantitative Description of Discontinuities in Rock Masses" 1978; W. S. Dershowitz and Einstein 1988; William S. Dershowitz and Herda 1992), defined as areas of intact unfractured rock where discontinuities have yet to propagate (de Vilder et al. 2017). Therefore, intact rock bridges could be defined as portions of intact rock separating joint surfaces (Elmo, Donati, and Stead 2018). Along the rock joint, the following are accounted: (1) rock bridge areas (intact rock), (2) open crack areas and (3) areas where rock bridges have already failed ("broken rock bridges") and where the joint surfaces are in

frictional contact. Figure 1.a presents a diagram of a discontinuity along which these three elements can be observed. The photo (Figure 1.b) was taken after the fall of an unstable block. The open crack areas as well as the broken rock bridges are visible. No rock bridges are observed in this photo; it is assumed that after the occurrence of the fall, there is no remaining rock bridge along the former joint. Before the fall of the unstable block, it can be expected that the broken rock bridges areas identified in the photo were in fact composed of intact rock and fresh intact rock rupture (broken rock bridges).

Conceptually, the location and distribution of rock bridges along a scar is supposed to control the failure mode (Tuckey and Stead 2016; Gregory M. Stock et al. 2011). For example, the presence of rock bridges over as little as just a few percent of the detachment surface is known to significantly increase the factor of safety by increasing apparent overall cohesion of a rock joint (Matasci et al. 2015; Tuckey and Stead 2016). Moreover, the location of a rock bridge is important for understanding if rockfall fails in tension or shear, as they can form a pivot point about which the failing rock block is able to potentially rotate

and fail in tension (Greg M. Stock et al. 2012; Bonilla–Sierra et al. 2015).

Previous research has shown that failure occurs through progressive fracturing of intact rock bridges, in a process termed step-path failure (Kemeny 2005; Eberhardt, Stead, and Coggan 2004; Scavia 1995; Brideau, Yan, and Stead 2009) that may in some cases be compared to a cascade-effect failure which can fail like dominoes along sloping channels (Bonilla–Sierra et al. 2015; Harthong, Scholtès, and Donzé 2012; Zhou et al. 2015). The contribution of rock bridges has been implemented in numerical

models of rock slope stability using apparent cohesion (Eberhardt, Stead, and Coggan 2004; Fischer et al. 2010; Gischig et al. 2011) or areas of intact rock (Stead, Eberhardt, and Coggan 2006; Sturzenegger and Stead 2009; Agliardi et al. 2013; Paronuzzi, Bolla, and Rigo 2016). These previous studies aim to analyse the failure modes and evolution of the rock bridges. However, they do not analyse the phenomenology of the rock bridges failure's propagation.

This paper studies (1) the phenomenology of the rock bridges failure propagation and (2) the influence of the rock bridges'

location, using a simple two-dimensional numerical model. It is structured as follow. In Section 2, the numerical modelling process considered in the study is presented: the geometry, characteristics and procedure of the models are defined. In Section 3, the results of our 2D simulations are shown: stresses redistribution along the joint after reducing the proportion of rock bridges is observed leading to the highlight of the rock bridges failure phenomenology. In Section 4, the results are discussed





and the influence of the rock bridges location, the role of the tensile shear strength on the phenomenology is evaluated. Finally, the conclusions are presented in Section 5.

## 2 Numerical modelling of the rock bridges failure propagation

The simulations were undertaken with UDEC (Universal Distinct Element Code), a two-dimensional distinct element code developed by Cundall (Cundall 1980) that can model the mechanical, hydraulic and thermal behavior of a fractured rock mass. This code has successfully been used to model the behaviour of rock discontinuities in past studies (Gu and Ozbay 2014; Jiang et al. 2006; He, Li, and Aydin 2018; Roslan et al. 2020). It has a scripting language embedded within it, FISH, that allows the user to create new model variables, customize functionality and interact with the model. This functionality has been decisive in the selection of the appropriate numerical tools, as it allows the rock bridge areas, open crack areas and broken rock bridges areas to be defined.

UDEC models the rock medium as a collection of blocks separated by joints regarded as smooth planes. The blocks can be rigid or deformable. They can mechanically interact through discontinuities. A distinction is made between data relating to blocks: nodes and corners, and data relating to discontinuities: contacts and domains. The characteristics of the discontinuities are defined through the contacts.

In order to study the phenomenology of the failure, an idealized two-dimensional numerical model has been defined. Therefore, there is no consideration of water infiltration, thermal implication or icing impact on the discontinuity at this stage, even though these phenomena may act as preparatory or triggering factors.

### 2.1 Geometry and definition of the two models

Two numerical models were built. Both models describe a potential plane failure along a pre-existing joint. The model 1 presents a joint with 80° dip angle while the model 2 presents a dip angle of 40°. These two models have been proposed in agreement with the objective of this work: to study the phenomenology of the rock bridges failure. To do so, a steeply dipping rock wall and a gentle slope are considered. These two cases are defined in function of the expected rockfall failure mode (shear or tension). It is expected that in the case of a steep slope, a tensile and/or shear failure mode will be observed. Indeed, authors (Greg M. Stock et al. 2012; Bonilla–Sierra et al. 2015) have highlighted that the location of a rock bridge is important for understanding if rockfall fails in tension or shear, as they can form a pivot point about which the failing rock block is able to potentially rotate and fail in tension. In the case of a gentle slope, only a shear failure mode is expected. Therefore, it is possible to assess the influence of the location of the rock bridges as well as the initial morphology of the rock wall.

The geometry of the two models is presented in Figure 2. The rock block presents a length of 6 meters and a width of 1,5 meters, leading to a total area of 9 m2, which, considering a out of the plane thickness of 1 meter, is also the volume (in m3) defined as "particularly dangerous for linear infrastructures and private residence" (Effendiantz et al. 2004). The total height





of the model is 12 meters. In practice, the geometry of the two models is the same; only the inclination of gravity is changed

(angle alpha on Figure 2).

During the meshing process, 128 contacts were created along the joint located between the block and the underlaying rock mass. Each contact can be defined by its coordinates in x and y (altitude). The behaviour of the rock joint is defined by the mechanical properties implemented for each individual contact (presented in the §2.2). As only contacts belonging to regions can be modified in UDEC, the rock joint was then divided into 100 regions of the same length that can represent either "rock

bridges" or "open crack areas". This division has been undertaken considering the FISH language. Each region can therefore include 1 or 2 contacts. During the computation process, the local stress distribution along the joint can lead to the rupture of some "rock bridges" regions, then becoming a region of "failed rock bridges" that behaves as an "open crack area". This phenomenon progressively increases the number of "open crack" regions along the joint.

Once the models are meshed, they have been loaded only by gravity to evaluate the initial local state of stress along the joint.

**2.2 Mechanical parameters**

An elastic model is assumed for the rock blocks and a Mohr-Coulomb elasto-plastic model is assumed for the rock joint (contacts along the joint). A contact exhibits a shear failure mode when the local stress reaches the Mohr-Coulomb failure criterion and a tensile failure mode when its tensile normal stress becomes equal to the assigned tensile strength.

The mechanical properties of the rock blocks (Table 1) were defined based on a literature review of a common limestone in

the French Alps ("Urgonien" limestone) (Frayssines 2005). This limestone has been considered as reference in this study as it forms high cliffs in South-Eastern France, where present traces of failed rock bridges are widely documented (Frayssines and Hantz 2006).

Along the rock joint, three types of contacts are considered:

1.    Rock bridges (RB) which behave elastically with the same characteristics as the intact rock. To determine the normal

115         and shear stiffness of the rock bridges, a centimetric opening of the joint has been considered;

2.    Open cracks (OC) which represent an absence of contact along the joint and behave in a perfectly plastic way;

3.    Rock bridges that failed due to stress transfers along the joint (RBF) and behave in a perfectly plastic way after their rupture.

RB and RBF have the same mechanical elastic parameters; the only difference between them comes from the fact that RB are

elastic while RBF are plastic.

The normal and shear stiffnesses of RB and RBF have been defined based on a literature review of Urgonien limestone fractures (Frayssines 2005). They are presented in Table 2.

The failure envelope properties of RB and RBF (cohesion, friction angle and tensile strength), were defined following a step-by-step procedure. As the objective of the numerical modeling is to study the phenomenology of the rock bridges failure

propagation, the failure criterion has to be close enough to the initial stresses along the joint, when considering only RB. Therefore, during a first step, the distribution of stresses has been evaluated and compared to "classical" failure criteria



provided in the literature (Frayssines 2005). Then, in a second step, the characteristics of the criteria have been decreased to fit the objective. The "classical" values and the ones defined with this procedure for the RB and RBF are presented in Table 3. Even if the values considered in the study are much lower than those found in literature, it is assumed that the failure propagation phenomenology will be the same as in reality. In the case of OC all the values are taken equal to 0 (Table 3).

## 2.3. Modelling protocol

The modeling protocol is based on the following steps. It is summarized in the Figure 3:

1. All the 100 regions and so the 128 contacts of the rock joint are initially considered as "rock bridges" (RB). In other words, 100% of the rock joint is defined as RB. The model is run to equilibrium under gravitational loading. This corresponds to the initial stage (Step 0);

2. Disturbances are introduced into the system. To do so, selected regions along the joint are transformed into "open crack" (OC) using FISH language (steps S1 to Sn, with n being the maximum number of steps before the block does not stabilize anymore). These regions can be selected randomly or chosen at specific locations. During these steps, X% of the rock joint is defined as OC and (100-X)% is defined as RB. At each of these calculating steps, the introduction of disturbance induces a stress redistribution along the joint, that leads to the failure of some rock bridges, then converted into RBF. This introduction of "open crack" areas simulate a virtual time as it represents the aperture of a crack and the propagation of the discontinuity through the rock bridges. It simulates the joint alteration that can be caused by, for example, water, freeze-thaw, root's growth, or another external parameter;

3. New "open crack" are introduced stepwise (step Sn) until the block does not stabilize anymore.

At each step of the modeling process, the following data is recorded:

- The normal and shear stresses at each contact along the rock joint,
- The number of contacts considered as open cracks (OP),
- The number of considered failed contacts (open crack and rock bridges that failed due to the increased of the stresses: OP + RBF).

This modelling protocol has the objective to analysis the rock bridge failure phenomenology. Based on this modelling protocol, different scenario have been considered:

- In scenario 1, the propagation of an open fracture was simulated. A 30 cm long area of open crack (OC) was initially defined, located at the lower part of the rock joint (near point A) for both models 1 and 2. Then, a progressive propagation of the open crack upwards was simulated (in this part of the study, contacts are not randomly modified from RB to OC.) At each step, the open crack area is enlarged.



- In scenario 2, the influence of rock bridges location along the joint was studied: (1) open cracks are introduced in the upper part of the rock joint (30 cm from point B), and (2) open cracks are introduced in the lower part of the joint (30 cm from point A). This protocol was followed for both dip angles of 40° and 80°.

- In scenario 3, 40 simulations with a random introduction of new OC were carried out to statistically compare results.

It can be noted that the numerical model has been validated by comparing the stresses evaluated by a simple theorical analytical calculation of a block laying on an inclined plane by numerical shear and normal stresses values.

## 3. Results

### 3.1. Stress transfer and RB failure induced by the introduction of new OC

To study the phenomenology of rock bridge failure (RB and RBF), the evolution of normal and shear stresses along the joint during the stepwise introduction of open cracks (OP) has been analyzed in detail. To do so, scenario 1 was considered.

Figure 4 presents, for both models 1 and 2, the distribution of the normal and shear stresses along the rock joint at different equilibrium steps S2 to Sn.

First, the distribution of the stresses along the rock joint is presented at Step S0, considering that the joint is only composed of rock bridges. In the case of model 1 (slope of 80 °), tension ($\sigma_n < 0$) is observed at the upper part of the block (near point B in the Figure 4a). In model 2 (slope of 40 °), no tension is observed.

For both models, at step 1, 10% of the rock joint is intentionally modified from RB to OC contacts. In both models, the introduction of OC results in a general increase of the shear stresses along the rock joint, with a stronger increase of these shear
stresses in the vicinity of the OC area. This increase in the shear stresses brings the joint closer to the failure criterion in the vicinity of the OC area, but elsewhere also, in particular at contacts located in the upper part of the rock joint (point B on Figure 4). The normal stresses slightly vary during this first stage.

During each subsequent step S2 to Sn, 2% of additional contacts are modified from RB to OC in the model 1 and 10% of additional contacts are modified from RB to OC in the model 2. These modifications induce the failure of some rock bridges
by increasing the shear stresses along the rock joint, but the model reaches a mechanical equilibrium at the end of each step. There is also an increase in the normal stresses along the rock joint. This phenomenon continues until the no mechanical equilibrium is reached anymore, which is associated with the downward sliding of the block (simultaneous failure of all the contacts).

The non-convergence of the model occurs when 16% of the contacts are converted to OC in the case of model 1, and 30% for
the model 2.

These results highlighted two phases during the rock bridges failure: a first phase during which only the intentionally created open cracks contacts are observed, and a second phase during which the stress transfers induce the additional failure of rock





bridges. In other word, in a first time, the crack enlarges without inducing rupture elsewhere, and in a second time the open crack reaches a state where rupture self-propagation starts until the block slides along the joint.

## 3.2. Rock bridges cascading failure phenomenology

To study more specifically this phenomenology, scenario 2 was considered. Results are presented in Figure 5 in terms of the proportion of so called "failed contacts" (OC + RBF) versus the proportion of OC along the joint. For both dip angles, there is a first linear phase during which the only "failed contacts" are the intentionally-introduced OC. During this first phase, the block remains stable, i.e. a mechanical equilibrium is reached after each introduction of new OC. Then, in a second phase, the redistribution of stresses caused by the introduction of new OC induces the rupture of some RB, that are converted into RBF. During this second phase, even a small increase in the proportion of OC leads to the rupture of additional rock bridges, which highlights the cascading failure phenomenology affecting the rock bridges. The slope of the linear regression in this second phase is around 7 in the case of model 1, and 3 in the case of model 2, meaning that the introduction of 1 OC leads to the failure of 7 RB for model 1, and 3 RB for model 2. This second phase starts for approximately 8% of the rock joint defined as OC for model 1 and 17% for model 2. The start of this phase differs slightly depending on the position of the RB and OC along the joint.

The non-convergence of the model starts when OC represents 17% and 27% of the joint for models 1 and 2 respectively.

Based on these preliminary results, scenario 3 was considered. Results are shown in Figure 6.

For both models 1 and 2, two phases in the propagation of the rupture may be identified for all the simulations carried out. In the case of model 1, the second phase starts for an average of $(8.5\pm1.5)$% of the rock joint defined as OC, and the slide of the block (non-convergence of the simulation) occurs for an average proportion of $(17.5\pm2.5)$%. Regarding the model 2, the second phase begins for an average of $(25\pm5)$% of the rock joint defined as OC, and the slide of the block occurs for an average proportion of $(35\pm5)$%.

## 3.3. Block displacement with time

In order to check whether there is a correlation between the two phases of rock bridges failure and the displacement that can be monitored on a potentially unstable block, a tracking point (C), shown in Figure 7, has been introduced. Such a point could easily be instrumented in the real case of motion tracking.

Scenario 3 was considered. The displacement of point C was studied versus the proportion of OC along the joint, which is a marker of "virtual time". The movement is no longer recorded as soon as all the contacts are failed, because the computation does not converge anymore.

Figure 7 shows that there is only one trend when considering the displacement. To be thorough, a smaller mesh has been defined, and the same results have been highlighted.





## 4. Discussion

The results highlight that the rock bridges failure phenomenology presents two phases: a first phase during which only the
intentionally created open cracks contacts are observed, and a second phase during which stress transfers induce the additional
failure of rock bridges. Based on these results, the influence of different parameters on this observed phenomenology was
tested. The results are presented below.

### 4.1 Influence of OC location on the evolution of RBF with time

As highlighted by different authors (Tuckey and Stead 2016; Gregory M. Stock et al. 2011), the location of the rock bridges
have a strong impact on the stability of a potential unstable block. To see whether our model leads to the same conclusion, the
following protocol has been followed:

1.  A number N of contacts is defined to be OC and randomly located along the joint. N is equal to 14 for the model 1
    (11% of the joint), and to 36 for the model 2 (28% of the joint). These values were chosen for the model to be at the
    end of the transition area and the beginning of the second phase where the cascading failure phenomenology affecting
the rock bridges is observed (section 3.2). As seen previously, these proportions are sufficient to induce the additional
    RBF;
2.  The number of considered failed contacts (OC+RBF) is determined;
3.  The number of failed contacts is compared to the average altitude of the OC contact.

Similarly, to scenario 3, 40 models are run.


The results are presented in Figure 8. Figure 8 (top part) shows the values of the minimum, maximum and average contact
altitude along the rock joint for both models 1 and 2. It also shows (bottom part) the total number of considered failed contacts
for a number N of contact defined to be OC with respect to the average altitude of the OC for both models 1 and 2.

Figure 8a presents the results of model 1. It highlights that there is a larger number of failed contacts (OC+RBF) when the OC
are localized on average in the upper part of the joint.

This difference highlighted between the two models can be explained by the distribution of the stresses along the joint. Indeed:

1.  In model 1, there is tension in the upper part of the rock joint (Figure 4a) when considering 100% of RB. To the
    contrary, in the model 2, there is no tension along the rock joint (Figure 4b);
2.  During the introduction of new OC, the stresses increase along the entire rock joint, and more specifically around the
OC area. The distance to the failure criterion must therefore plays an important role if it is assumed that the increase
    in stresses is done in a homogeneous way, which seems to be the case based on the Figure 4. For model 1, the distance
    to the criterion is the smallest in the upper part of the block, and vice versa for model 2, which may explain the
    influence of the position of the open crack.





The Figure 8 highlights the presence of critical position of the OC area. The critical position could be defined as the position
where, for a same proportion of OC, more RBF will be generated than in any other position along the joint. In the case of the
model 1, the critical position of the open cracks area corresponds to the upper part of the joint (i.e. RB located preferentially
in the lower part of the joint). To the contrary, for model 2, it is the OC area located in the lower part of the joint that corresponds
to the critical position (RB located in the upper part of the joint). This results combined with geophysical tools investigations
(Gregory M. Stock et al. 2011; Matasci et al. 2015; Paronuzzi, Bolla, and Rigo 2016; Guerin et al. 2019; Frayssines and Hantz
2006; Paronuzzi and Serafini 2009; Spreafico et al. 2017), could allow to prioritize the potentials unstable blocks

**4.2 Role of the tensile strength on the evolution of RBF with time**

On the presented study, only shear failure was considered with the Mohr-Coulomb envelope and tensile failure was
disregarded. This assumption is debatable in comparison with reality and will be discussed hereafter.

In order to study the role of introducing a tensile strength, a new model 3 has been defined and run. It is based on model 1 (dip
angle of 80 °) as model 1 shows tension. In the new model, a tensile truncation was added to the Mohr-Coulomb failure
criterion. The tensile strength TS has been taken equal to the Uniaxial Compressive Strength UCS value divided by 10
(UCS/10). The compressive strength is calculated according to Eq. (1).

$$\text{UCS} = \frac{2c\,cos\varphi}{1-sin\varphi}, \tag{1}$$

with $c$ being the cohesion and $\varphi$ the friction angle.

The mechanical characteristics of the model 3 are listed in Table 4. The cohesion value has been increased in comparison to
Model 1 for numerical modelling requirements: when considering the same cohesion value, the model was not converging.
The cohesion value has been increased until the model could be run.

The results are presented on Figure 9. The tensile truncation of the Mohr-Coulomb failure criterion results in tensile failure of
6% of the joint at the initial step S0. At each subsequent step, 10% of additional OC are introduced along the rock joint.
Because the cohesion is three times higher than for model 1, the stresses along the joint are further away from the failure
criterion of rupture than for model 1 (Figure 09). As observed previously, the normal and shear stresses progressively increase.
It can be noted, as for the previous models, a more significant increase of the shear stress in the vicinity of the OC area. Up to
40% of the joint can be defined as OC before the calculation does not converge anymore.

For model 3, the transition phase identified on Figures 5 and 6 is comprised between 40% and 50%, while in model 1, it is
comprised between 8% and 15%. In other words, when increasing cohesion value, the proportion of open crack needs to be
higher to reach the cascading failure affecting the rock bridges than when considering low cohesion value. It justifies that in
reality, as the cohesion value of the rock bridges are 500 times higher than in the study presented in this paper, only a few
portions of rock bridges allow a potential instable block to be in place. The second phase observed in the paper occurs instants
before the fall of the block.






This study shows that, when considering tensile failure though the tensile truncation of the Mohr-Coulomb failure criterion, a proportion of failed rock bridges comes from the tensile stresses along the joint. However, the same "bi-phase" propagation failure phenomenology was observed regardless the comprehensive consideration of the tensile failure.

### 4.3 Influence of RBF's shear strength on the results

In the modelling procedure presented in §2.3 and applied to models 1 to 3, the rock bridges that failed during the calculation (RBF) are considered to keep the same shear strength values as RB. This hypothesis has been made to consider asperity that can exists along areas of failed rock bridges. An alternate approach would be to consider that RBF behave as OC. This is discussed hereafter, by the mean of an additional model 4 comprising only two types of contacts: RB and OC. RBF are considered to behave as OC. This model is based on model 1 (dip angle of 80 °), to which it will be compared.

The new OC will be introduced in the upper part of the joint as it has been highlighted that for model 1, there is a larger number of failed contacts (OC+RBF) when the OC are localized on average in the upper part of the joint.

Figure 10 presents the distribution of stresses along the joint at different steps of computation for models 1 and 4. The first OC area is introduced in the upper part of the joint, 10 cm away from point B. It is observed, as previously, a general increase in shear stresses and a very small increase in normal stresses. Model 1 stops converging when 18% of the joint is defined as OC

(Figure 10a), which is in agreement with what was observed before. When considering 16% of joint defined as OC (last step before the model does not converge), there is 22% of failed contacts (OC+RBF). Model 4 stops converging for 26% of OC (Figure 10b). Therefore, considering 2 or 3 types of contacts gives similar results.

### 5. Conclusions

The aim of the work presented in this paper is to study the phenomenology of the rock bridges failure. To do so, a block sliding

along an inclined plane has been considered and modeled through Discrete Element Modeling (DEM). It has been assumed that the discontinuity (rock joint) separating the block from the inclined plane is composed of (1) rock bridges (RB) (portion of intact rock separating joint surfaces) and (2) open crack (OC) (no contact between the block and the plane). Rock bridges can fail due to a change in the shear and normal stresses along the joint (RBF). Because of the phenomenological goal of this work, it is assumed that the shear strength characteristics (cohesion and friction angle) of the rock bridges are low (see Table

3) in comparison with the values classically considered as representative of a rocky environment. This strong assumption as the failure criterion has to be close enough to the initial stresses along the joint.

The modelling protocol implemented allowed the following observations to be highlighted:

- The introduction of OC and the failure of rock bridges in the vicinity of OC areas leads to a stress redistribution along
the rock joint. In particular, there is an increase in the shear stress in the vicinity of the OC areas, that can lead to the failure of the neighboring RB;
- There are two phases in the propagation of the rock bridges failure:



- o a first phase during which no rock bridges failure is induced. During this first stage, only the OC contacts are at failure;

- o a second phase, during which even a small increase in the proportion of OC leads to the rupture of additional rock bridges, which highlight the cascading failure phenomenology affecting the rock bridges. This phenomenon continues until the block slides or tilts. In the case of a steep slope (80 ° - model 1), a 1% increase in the proportion of OC leads to the failure of 6% of RB (ratio of 7). For a gentle slope (40 ° - model 2), the ratio is equal to 3;

- The study of displacements does not make it possible to identify the two phases in the failure propagation. It highlights that there is only one trend when considering the displacement. Therefore, the study of displacements does not make it possible to distinguish between the two, previously described phases of crack propagation;

- The position of the RB plays an important role in the stability of the block. In the case of a steep slope, the critical position of the open cracks area corresponds to the upper part of the joint (i.e. RB located preferentially in the lower

part of the joint). To the contrary, for a gentle slope, it is the OC area located in the lower part of the joint that corresponds to the critical position (RB located in the upper part of the joint). This result is correlated with the presence of tension in the upper part of the rock joint in the case of the model 1.

The observation made through numerical models about the cascade effect is an interesting result as it leads to a better understanding of the failure mechanism leading to the triggering of a rockfall. It helps complement the current assessment

methods of the failure probability of the rockfall hazard. In particular, it describes why it can be so challenging to assess the occurrence probability of such events.

Moreover, the work presented in this paper highlighted the importance of the rock bridges location and their assessment. Therefore, the use of geophysical investigations could allow to prioritize the potentials unstable blocks.

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

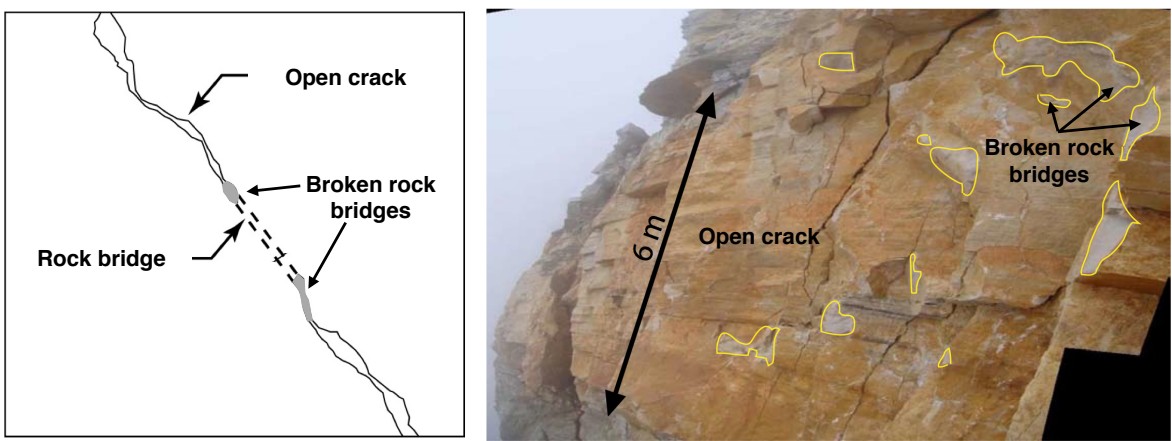

**Fig. 1a (left) and b (right). Definition of rock bridges, open crack and failed rock bridges areas. Modified from Levy (C. Levy 2011) (with reproduction authorization).**

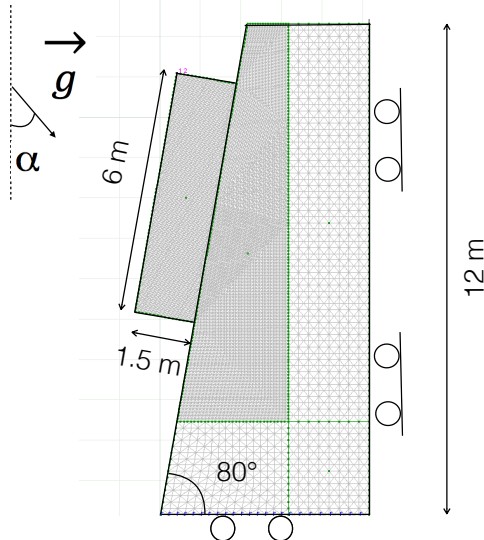

**Fig. 2. Geometry of the both models. α is equal to 0° for the model 1 (slope angle: 40°) and is equal to 40° for the model 2 (slope angle: 80°).**


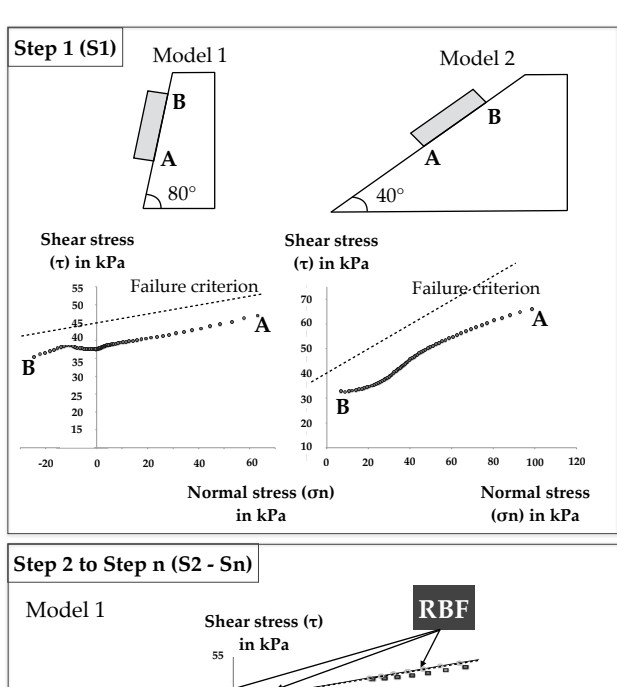

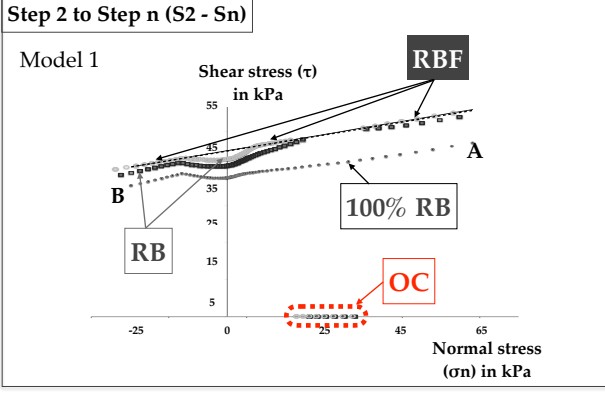


**Fig. 3. Modelling process.**

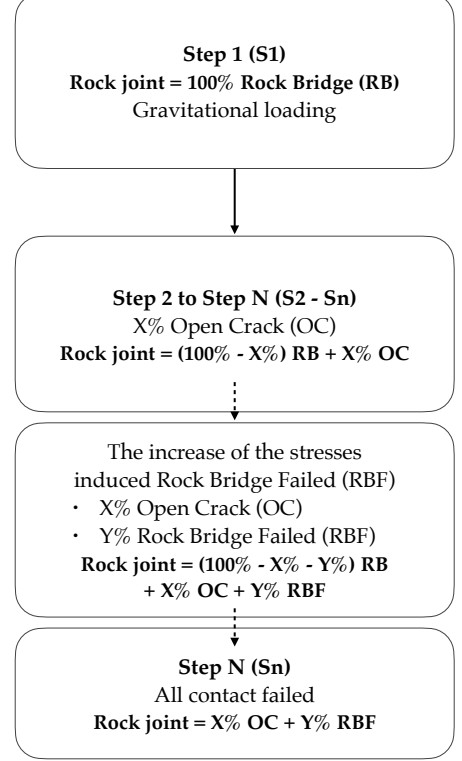
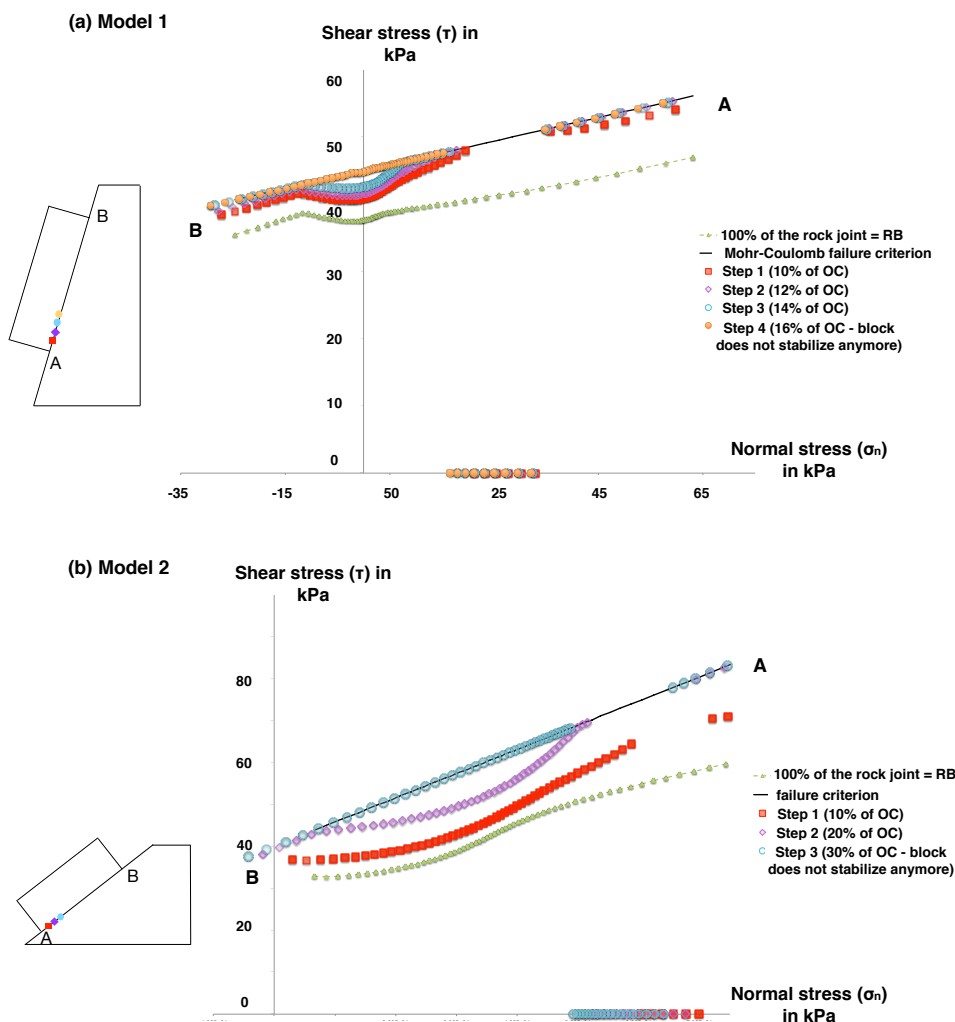

**Fig. 4. Distribution of the normal and shear stresses for models 1 and 2. The different steps represent the introduction of new OC until the model does not converge anymore. The points on the x-axis have normal stress, but no shear stress as if the friction angle was zero. Each colour between point A and point B in the model corresponds to the step presented in the graph.**





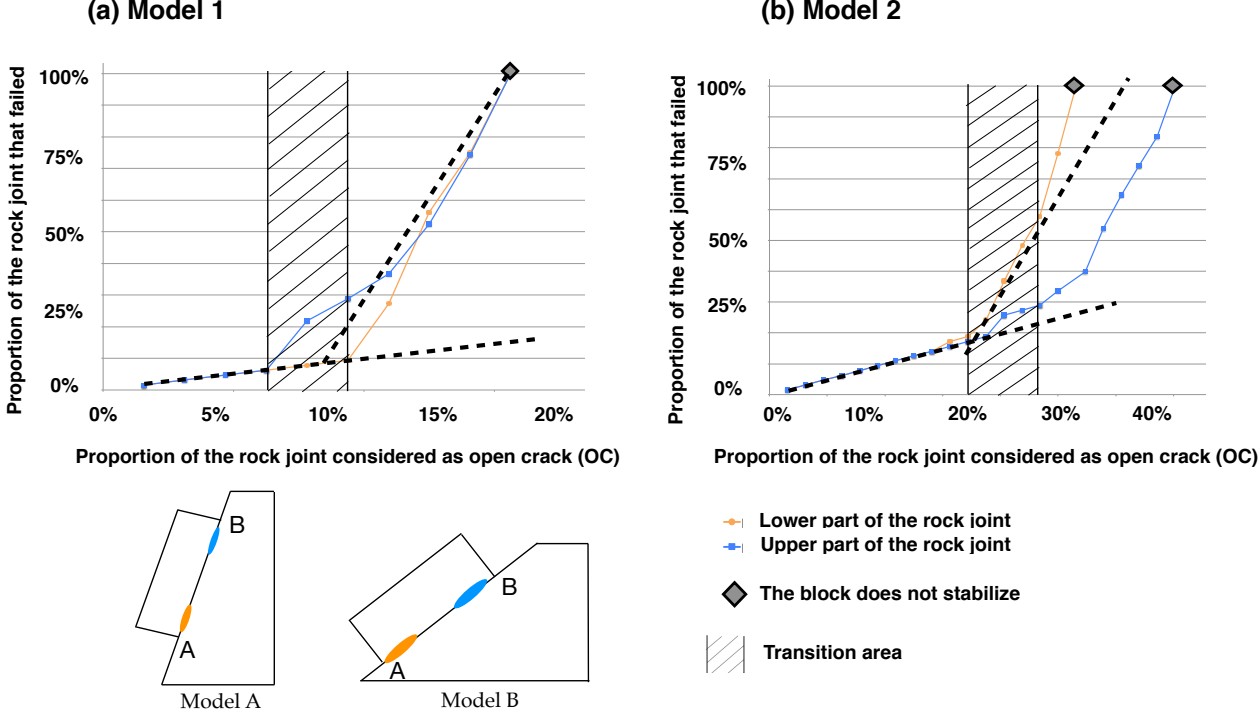

**Fig. 5. Propagation of rock bridge failure for models 1 and 2. Intentionally introduced OC are located in the upper part of the joint (blue curve) or in the lower part of the joint (orange curve). The proportion of the rock joint that failed is defined as a ratio between the number of failed contacts (OC + RBF) and the total number of contacts (OC + RB + RBF).**


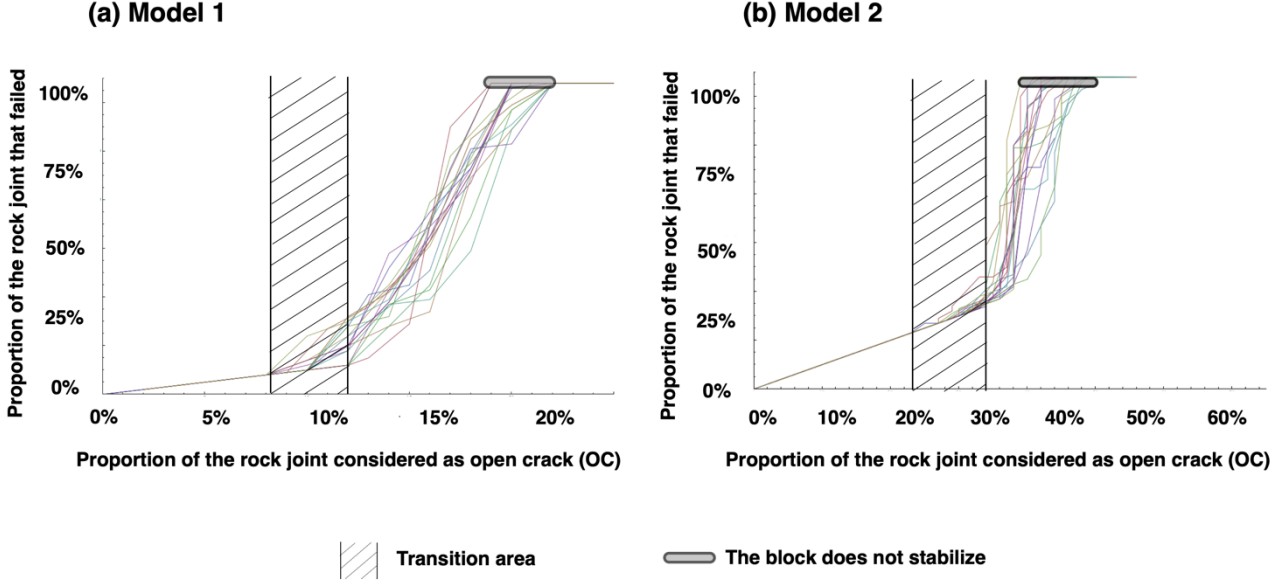

**Fig. 6. Propagation of rock bridge failure for models 1 and 2 in the case of randomly introduced new OC.**




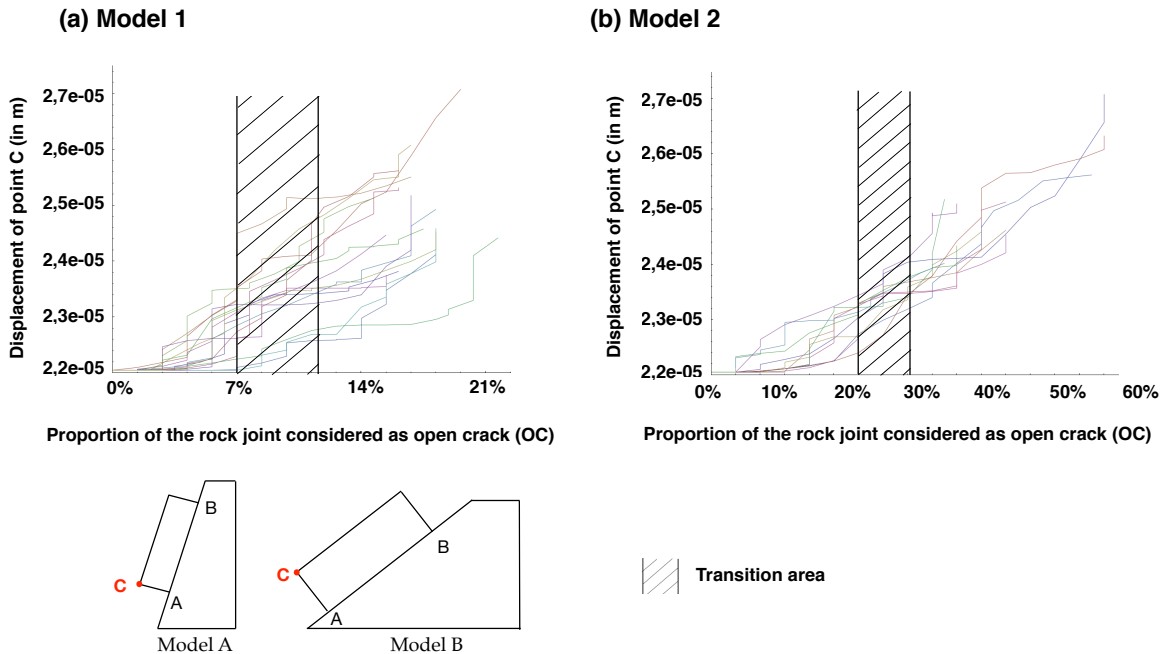

**Fig. 7. Displacement of point C (in meters) with respect to the proportion of OC along the joint, for models 1 and 2. The**
**transition zone presented here corresponds to the one defined previously (§3.2).**

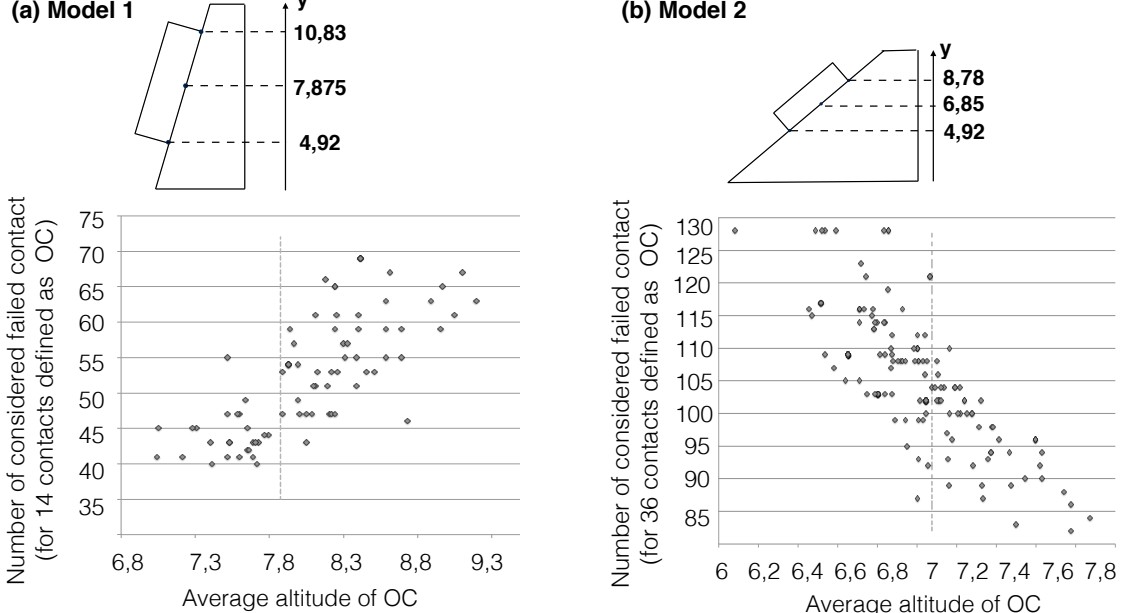

**Fig. 8. Number of considered failed contacts for a number N of contact defined to be OC with respect to the average**
**altitude ("y" coordinate) of the OC contacts, for models 1 and 2. N is equal to 14 and 36 respectively for model 1 and 2.**





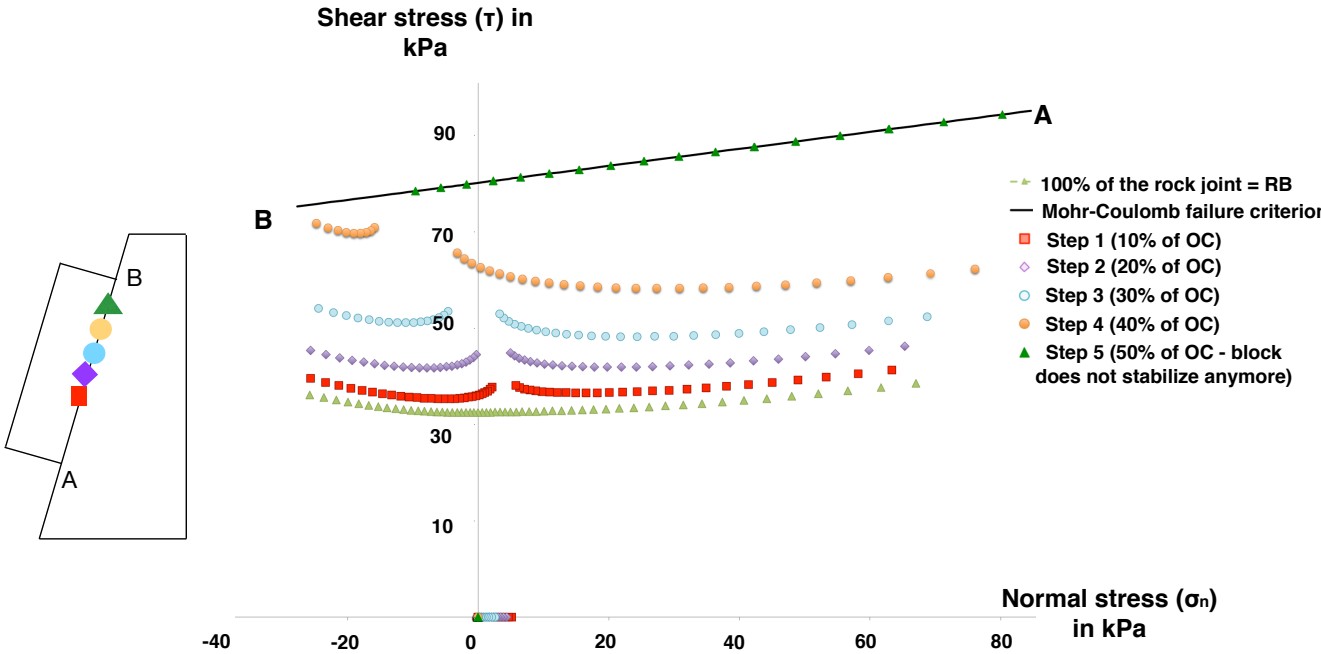

Fig. 09. Distribution of the normal and tangential stresses in the plan of Mohr for the model 3. The various steps represent each time the introduction 10% of open cracks (OC), until the non-convergence of the model.




**(a) Model with rock bridges that failed (RBF) considered keeping the same mechanical properties**

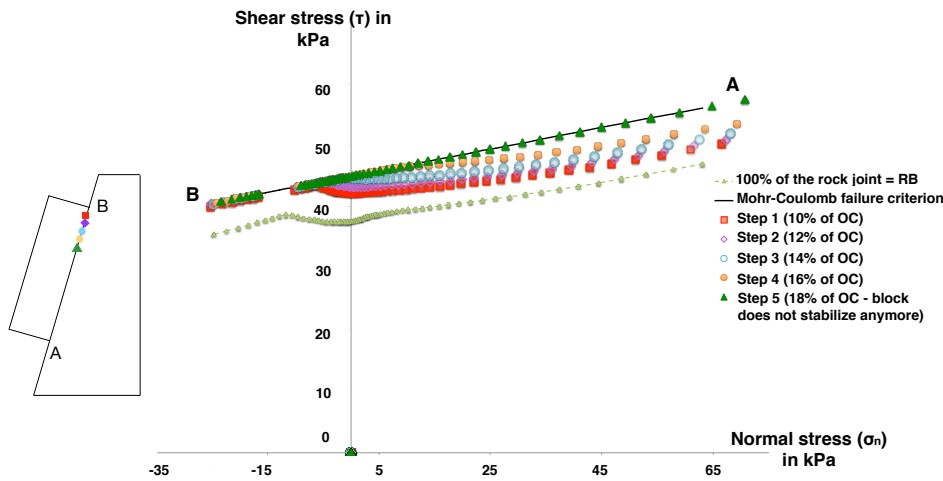

**(b) Model with rock bridges that failed (RBF) considered as open crack (OC)**

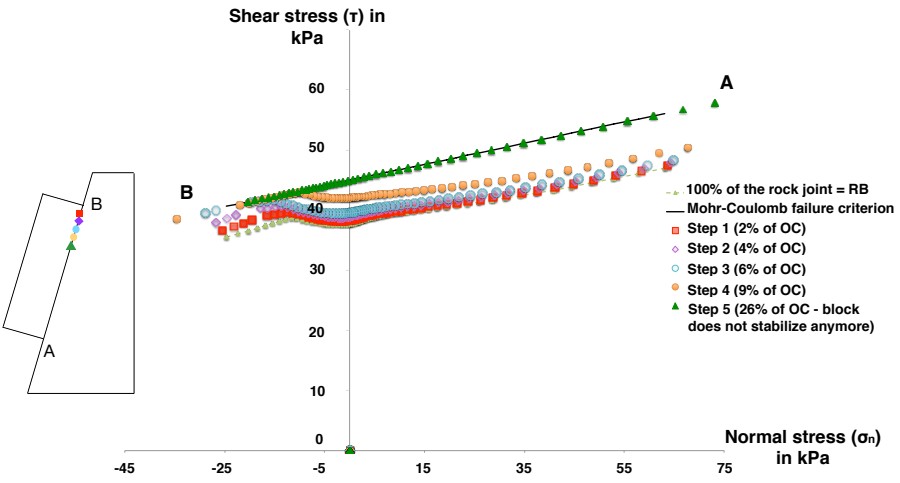

Fig. 10. Normal and shear stress distribution for (a) model 1 in the case where 3 types of contacts are considered (open crack (OC) – rock bridges (RB) – rock bridges that failed (RBF)) and (b) model 4 if the contacts defined as rock bridges that failed are automatically changed to open crack (OC).

Table 1. Mechanical properties of the rock mass based on Urgonien limestone.

| Young's modulus (E) | Poisson's ratio (ν) | Density (ρ) |
|---|---|---|
| 68.9 GPa | 0.31 | 26.9 kN/m³ |






**Table 2. Elastic mechanical properties of typical rock joints in Urgonien limestone.**

| Normal stiffness ($k_n$) | Shear stiffness ($k_s$) |
|---|---|
| 6.9 GPa/m | 2.7 GPa/m |

**Table 3. Shear strength characteristics of RB, RBF and OC areas along the joint for both models A and B.**

| | "Classical" Rock bridges characteristics | "Rock bridges" (RB) and Failed Rock Bridges (RBF) Model 1 | "Rock bridges" (RB) and Failed Rock Bridges (RBF) Model 2 | "Open cracks" (OC) Model 1 and model 2 |
|---|---|---|---|---|
| **Cohesion C** | 23 MPa | 45 kPa | 40 kPa | 0 Pa |
| **Angle of friction** | 54° | 10° | 30° | 0° |
| **Tensile strength TS** | 7 MPa | 10 kPa | 10 kPa | 0 Pa |

**Table 4. Mechanical characteristics of rock bridges in the model 3 used when studying the effect of tensile strength. The dip angle is equal to 80 °.**

| | "Rock bridges" (RB) Model 1 | "Rock bridges" (RB) Model 3 |
|---|---|---|
| **Cohesion C** | 45 kPa | 130 kPa |
| **Friction angle** | 10° | 10° |
| **UCS** | 107 kPa | 312 kPa |
| **Tensile strength TS** | 10 kPa* | 31,2 kPa |

* as defined in model 1