# Peer review of "Cascade effect of rock bridge failure in planar rock slides: explicit numerical modelling with a distinct element code"

_Natural Hazards and Earth System Sciences, 2020_

## Referee Comment (RC1) · Anonymous Referee #1 · 11 Nov 2020

The paper deals with a topic which is of great interest to those involved in rock collapses and rockfalls: the influence of the rock bridges location on the shear and tensile strength along a discontinuity plane, as well as the associated phenomenology of rock bridge failure propagation, which were analysed through a two-dimensional distinct element numerical modelling.

In my opinion, the weakest point of the paper is its very poor connection with real case conditions. In particular, the selected strength properties of rock bridges, as well as those associated to open cracks (OC) are extremely low. This results in a high percentage occurrence of rock bridge with low strength, which is not representative of

real conditions, where we have a much lower occurrence of high strength rock bridges. About this, at line 129 Authors state that "Even if the values considered in the study are much lower than those found in literature, it is assumed that the failure propagation phenomenology will be the same as in reality". I personally disagree with this assumption, however I believe that it should be extensively discussed and adequately justified by Authors.

In addition, I found some inconsistencies and unclear point throughout the manuscript, therefore I recommend Authors performing an accurate revision before its acceptance, based also on the following specific comments:

L 26-32: Where Authors list the main methods for failure probability assessment. Please consider adding the kinematic analysis method.

L 33: "The main parameter. . .are", please check correct conjugation

L 35 and elsewhere: "W. S. Dershowitz and Einstein 1988". Please carefully check references and in-text citations according to journal instructions

L102-103: " During the computation process, the local stress distribution along the joint can lead to the rupture of some "rock bridges" regions, then becoming a region of "failed rock bridges" that behaves as an "open crack area"". Actually, if I correctly understood, for most of the models, the mechanical properties of failed rock bridges are the same as those of intact rock bridges (see table 3). Indeed, it is not clear what is the difference between RBF and RB in terms of behaviour within the model. RB have the same parameters of RBF, thus the behaviour of both should be defined as elasto-plastic (while in Line 120 RBs are defined purely elastic).

L130: "In the case of OC all the values are taken equal to 0 (Table 3)": please justify this fundamental choice, since in my opinion, especially when talking about shear resistance, the condition of zero strength along a discontinuity is impossible to reach.

L135 (Step 0): the initial step in Figure 3 is called S1 (not step 0), and the following

steps and figures 4, 9, 10 disagree too; please standardize the step numbering in the text and figures

L151-152: "This modelling protocol has the objective to analysis the rock bridge failure phenomenology. Based on this modelling protocol, different scenario have been considered" – please check english grammar.

L153-155: The introduction of a 30 cm long open crack corresponds to 5% of the total discontinuity length. Figure 4 starts from 10% of OC (which corresponds to 60cm), and in figure 4a 2% step are analysed (which correspond to OC increase along the discontinuity of 12 cm). Please clearly explain in the text the exact new OC length added at each step for the three scenarios.

L157: please, explain the difference (if any) between scenario 1 and scenario 2 (2) (open cracks introduced in the lower part).

L169: "steps S2 to Sn" - see previous comment L135: please standardize the step numbering in the text and figures

L179-180: "These modifications induce the failure of some rock bridges by increasing the shear stresses along the rock joint". It could be interesting to add a figure to see which rock bridges fail with respect to the introduced OC contacts.

L184: 84% and 70% occurrence of rock bridge at failure in my opinion is too high (see general comment).

L198: if Authors refer to the slope of the steepest dashed line, I would say a rate of at least 10 (instead of 7) and 5 (instead of 3) for model 1 and 2 respectively.

L199: Authors should describe here the meaning of the "transition area" presented in figures 5-7

L199-202: the OC values of 8% and 17% corresponding to the beginning of the second phase and 17% and 27% corresponding to the non-convergence of the model seem to

not completely fit with figure 5 axes labels. I am wondering if some mismatch occurred between the axes ticks and labels.

L216-217: this sentence is unclear, please explain better

L225-233: I'm not sure to have correctly understood this point. By looking at Fig. 6, for %OC = 11% I expect a maximum proportion of the failed contacts of 25% (which corresponds to about 32 contacts). Why is the number of failed contacts between 40 and 70 in figure 8a? I have the same doubt for model 2.

L234: If 40 models were run, why do figure 8a and 8b report a higher number of points? (at least 70)

L240: Please comment results also for model 2.

L245 "must therefore plays", please correct typo.

L255 please correct to "potential unstable blocks"

L258 if tensile strength is not considered in the previous analyses, why is it reported in table 3?

L265 and table 4: This continuous change of strength parameters is not scientifically sound, and it should be supported by a stronger discussion. Please explain why did Authors not choose a unique set of strength parameters for all the models

L269: How do Authors calculate this 6% value? Is it related to the number of green triangles with normal stress lower than -10kPa? Please explain

L271 and Figure 9 caption, please correct figure 09 to figure 9

L277-278: "It justifies that in reality, as the cohesion value of the rock bridges are 500 times higher than in the study presented in this paper, only a few portions of rock bridges allow a potential instable block to be in place". I agree with this statement; why did not author decide to use more realistic strength (and rock bridges portion) values?

(see also general comment)

L281: please correct "though" to "through"

L285-287: from a geomechanical point of view it is not correct to consider failed rock bridges to have the same strength of intact rock, as well as it is not correct to consider them to have zero strength. Why did Authors not consider employing a continuously yielding model?

L295-297: Probably it could be interesting to see the propagation of rock bridge failure (like in figure 5) for model 4

Figure 2. The gravity direction is in my opinion misleading, I suggest adding the 40° slope model. Moreover, it seems to me that the alpha associations in figure captions are wrong, i.e. alpha is zero for 80° (not 40°) slope angle and 40° for 40° (not 80°) slope angle.

Figure 3. Expand the caption to better describe the figure.

Figure 5 and figure 7: please put the model 2 sketch on the right side, under the corresponding graph.

---

## Referee Comment (RC2) · Anonymous Referee #2 · 16 Nov 2020

Comments on "Cascade effect of rock bridge failure in planar rock slides: explicit numerical modelling with a distinct element code" by Delonca et al. submitted to the journal of NHESS. The authors attempt to estimate the cascade effects of rock bridge failure via DEM. The paper is interesting, but it falls on the borderline regarding the scope of NHESS from my personal point of view. So some more application or relationship regarding to principal/traditional aspect of NHESS should addressed. Some specific comment listed bellows:

1. Why only two aspect /gravity directions from to the same mesh been analyzed. This phenomenon represent to two different sliding behaviors fall vs. slide. But, what about

the behavior in transition zone? It should be address to judge the research design of this manuscript/study. 2. Line 274-275, " the transition phase identified in Figs 5 and 6. . . and 15%" this sentence not clear identify on the figures or the manuscript. 3. The conclusions section: the section should be rewrite to extract the conclusions, not extract directly from some sentences in each subsection again. 4. Fig 5 and 7, typos in Model A and Model B ? Should it be Model 1 and 2? 5. Some limitations should be addressed, e.g. tensile cracks, shear cracks, asperity etc. vs. planner open cracks simplified in this study.

---

## Referee Comment (RC3) · Anonymous Referee #3 · 23 Nov 2020

Dear Authors I read your paper with great interest. Rock bridges are indeed a "hot topic," and your numerical approach should interest the scientific community. However, I have some comments and some questions that need to be addressed before publication. The other reviewers have already put some forwards, but I have some more.

1) Why is your contact number different from your region's number? What kept you from maintaining them equal? I understand that 100 is a "round number" but that some regions may have more than one contact affects clarity.

2) What random distribution was used to select the regions to transform in OC? Was the relative location of the different regions considered in the randomization? Why did

you choose this specific randomization method? How was "10% of the rock joint is intentionally modified from RB to OC contacts"? Is it again through a randomization tool? Of what kind?

3) Is there any account of elastic rebound in your model immediately after a rupture?

4) I also have some issues with the statement in line 211-212 "such a point could be easily be instrumented in the real case of motion tracking" – I have doubts about that, at this scale, since you are modelling displacement of less than a hundredth of a millimeter, and this quantity is really difficult to monitor on field. Moreover, what do you mean by "Figure 7 shows that there is only one trend when considering displacements" (line 216)?

5) Finally, the limitations of 2D approaches to study rock bridges should be discussed. We know that in 2D rock bridges are scale dependent (Elmo, 2018). Do you believe that the reduction of the strength parameters you imposed was scaled accordingly with your geometry? How would a tri-dimensional geometry affect your simulations?

Specific comments

- The scheme in fig 1a is not clear and readable; please emend it maybe using colors

- Why is g horizontal in figure 2? Please show the mesh of your model.

- Fig 3 caption must be way more informative. The graphs on the left should be described, especially the Step 2 one where we see two evolutions (?) one in black and one in gray

- The caption of Fig 4 should include the fact that it refers to Scenario 1, as Scenario 2 in the caption of Fig.2

- In Fig. 5 and fig 6 maybe substitute "the block does not stabilize" with "collapse" or "non-convergence of the simulation"

- Fig. 8. Wouldn't it be better to plot the height (better than altitude) of OC using the un-

[Figure]

stable block's base as reference? So a length that spans between 0 and 6*cos(alpha)? In this way, it would be easier to compare the average height of the contacts with the block's median point. Why do you think in model 2 the average height of the OC that induce collapse is higher (7 m) than the median poinf of the block? And why they seem to coincide in model 1? This needs to be discussed. It would also be useful if you plot a histogram of the average heights. Finally, the unit of measures [m] should be indicated in the figure.

- Fig 9 (not 09) – Scenario 3, not model 3

- In general, for many figures, the unit of measurement should be indicated in [] brackets instead of "in kPa"

L 23 – in other words

L 33 – are rock bridges

L 100 – "This division has been undertaken using FISH"

L 245 – play

L 249 – Figure 8 (no "the")

Check reference format: Brideau, Delonca (journal name abbreviated), Spreafico
* * *

---

## Author Comment (AC1) · 30 Dec 2020

Santiago, Chile, January 2021

**Authors' Response**

**Nhess- 2020-279 - Cascade effect of rock bridge failure in planar rock slides: explicit numerical modelling with a distinct element code.**

Dear Editor,

The authors thank the 3 reviewers for their exhaustive work and their valuable comments. They have allowed us to significantly improve the quality of our paper and to better highlight the relevance of our work. In the following we present a response to their comments. Moreover, we will submit a revised version of the manuscript, with the modifications highlighted in the red color.

**Comments of Reviewer 1**

**General comments:**

The paper deals with a topic which is of great interest to those involved in rock collapses and rockfalls: the influence of the rock bridges location on the shear and tensile strength along a discontinuity plane, as well as the associated phenomenology of rock bridge failure propagation, which were analysed through a two-dimensional distinct element numerical modelling.

In my opinion, the weakest point of the paper is its very poor connection with real case conditions. In particular, the selected strength properties of rock bridges, as well as those associated to open cracks (OC) are extremely low. This results in a high percentage occurrence of rock bridge with low strength, which is not representative of real conditions, where we have a much lower occurrence of high strength rock bridges. About this, at line 129 Authors state that "Even if the values considered in the study are much lower than those found in literature, it is assumed that the failure propagation phenomenology will be the same as in reality". I personally disagree with this assumption, however I believe that it should be extensively discussed and adequately justified by Authors.

**Response:** First of all, the author would like to thank the reviewer for this very relevant comment. The distant connection with real case conditions has been the subject of an internal discussion between the authors of this paper. Low rock bridges mechanical properties have deliberately been chosen here in agreement with the objective of this study, which is to highlight the phenomenology of rock bridge failure propagation, and not to accurately represent the behaviour of rock bridges themselves. Indeed, several authors (e.g.[1]–[3]) have highlighted the really low proportion of rock bridge existing right before the fall (between only 0.2 to 5% of the detachment surface). In particular, Frayssines & Hantz (2009) have shown that rock blocks could remain stable for a long time thanks to rock

bridges and that the rock bridge proportions in the failure surfaces in these cases can be very small (less than 1% of the detachment surface). This highlights the difficulty of studying failure propagation, as a rock block can remain stable even with a really low rock bridges proportion, as well identified and reminded by the reviewer. From a numerical point of view, modelling less than 1% of the joint as rock bridges would require an extremely dense meshing, due to the high stress concentration and stress gradients in the rock bridge areas. This may be the topic of a future study.

Moreover, previous research has shown that failure occurs through progressive fracturing of intact rock bridges, in a process termed step-path failure [4]–[7] that may in some cases be compared to a **cascade-effect failure** as they can fail like dominoes along sloping channels [8]–[10]. To study this cascade-effect failure, the focus of this paper is the phenomenology and not the values of the parameters, even if parallels between past studies and our work is intended to be done.

We feel confident in our methodology as other calculations have been performed with higher mechanical properties and have shown a similar phenomenology. An example of this is the model 3, which considers higher cohesion value (130 kPa for model 3). A detailed discussion regarding this subject has been added in the revised version of the paper (4.4 Influence of the rock bridges mechanical properties).

Finally, we are well aware that this study is a preliminary study of a complex topic, and while there is an element of arbitrariness in the choices made, the results allow us to highlight some interesting results.

Specifics comments:

In addition, I found some inconsistencies and unclear point throughout the manuscript, therefore I recommend Authors performing an accurate revision before its acceptance, based also on the following specific comments:

| Specific comments                                                                                                                                         | Response                                                 |
|-----------------------------------------------------------------------------------------------------------------------------------------------------------|-----------------------------------------------------------------|
| L 26-32: Where Authors list the main methods for failure probability assessment. Please consider adding the kinematic analysis method.                    | Kinematic analysis method has been added [11]–[13].             |
| L 33: "The main parameterare", please check correct conjugation                                                                                           | Done in the revised manuscript                                  |
| L 35 and elsewhere: "W. S. Dershowitz and Einstein 1988".
Please carefully check references and in-text citations
according to journal instructions | The entire manuscript has been revised regarding the references |

|  <li>L130: "In the case of OC all the values are taken equal to 0 (Table 3)": please justify this fundamental choice, since in my opinion, especially when talking about shear resistance, the condition of zero strength along a discontinuity is impossible to reach.</li> <li>L135 (Step 0): the initial step in Figure 3 is called S1 (not step 0), and the following steps and figures 4, 9, 10 disagree too; please standardize the step numbering in the text and figures.</li> <li>L151-152: "This modelling protocol has the objective to analysis the rock bridge failure phenomenology. Based on</li>  |
|----------------------------------------------------------------------------------------------------------------------------------------------------------------------------------------------------------------------------------------------------------------------------------------------------------------------------------------------------------------------------------------------------------------------------------------------------------------------------------------------------------------------------------------------------------------------------------------------------------------------------|
| L135 (Step 0): the initial step in Figure 3 is called S1 (not
step 0), and the following steps and figures 4, 9, 10 disagree
too; please standardize the step numbering in the text and
figures.Step 0 correspond to 100% of the joint as RB.
has been added to Fig 4, 9 and 10. Fig 3 h
been modified.L151-152: "This modelling protocol has the objective to
analysis the rock bridge failure phenomenology. Based onThese sentences have been rephrased.                                                                                                                                              |
| L151-152: "This modelling protocol has the objective to analysis the rock bridge failure phenomenology. Based on                                                                                                                                                                                                                                                                                                                                                                                                                                                                                                           |
| this modelling protocol, different scenario have been considered" – please check english grammar.                                                                                                                                                                                                                                                                                                                                                                                                                                                                                                                          |
| L153-155: The introduction of a 30 cm long open crack corresponds to 5% of the total discontinuity length. Figure 4 starts from 10% of OC (which corresponds to 60cm), and in figure 4a 2% step are analysed (which correspond to OC increase along the discontinuity of 12 cm). Please clearly explain in the text the exact new OC length added at each step for the three scenarios.                                                                                                                                                                                                                                    |
| L157: please, explain the difference (if any) between In the case of the lower part, both scenarios a scenario 1 and scenario 2 (2) (open cracks introduced in the lower part). In the case of the lower part, both scenarios a the same, except for the first step. The difference has been clarified with the previous comment in the same set of the lower part.                                                                                                                                                                                                                                                        |
| L169: "steps S2 to Sn" - see previous comment L135: Authors have carefully reviewed the paper order to standardize the step numbering in the text and figures                                                                                                                                                                                                                                                                                                                                                                                                                                                              |
|                                                                                                                                                                                                                                                                                                                                                                                                                                                                                                                                                                                                                            |

| joint". It could be interesting to add a figure to see which rock bridges fail with respect to the introduced OC contacts.                                                                                                                                                                                            |                                                                                                                                                                                                                                                                                                                                                                                                              |
|-----------------------------------------------------------------------------------------------------------------------------------------------------------------------------------------------------------------------------------------------------------------------------------------------------------------------|--------------------------------------------------------------------------------------------------------------------------------------------------------------------------------------------------------------------------------------------------------------------------------------------------------------------------------------------------------------------------------------------------------------|
| L184: 84% and 70% occurrence of rock bridge at failure in my opinion is too high (see general comment).                                                                                                                                                                                                               | The authors agree with this comment. However,
as explained, we are studying the
phenomenology and not a specific realistic case.
A comment has been added in the conclusion
section.                                                                                                                                                                                                             |
| L198: if Authors refer to the slope of the steepest dashed line, I would say a rate of at least 10 (instead of 7) and 5 (instead of 3) for model 1 and 2 respectively.                                                                                                                                                | Ok.                                                                                                                                                                                                                                                                                                                                                                                                          |
| L199: Authors should describe here the meaning of the "transition area" presented in figures 5-7                                                                                                                                                                                                                      | A comment has been added at the end of §3.2.                                                                                                                                                                                                                                                                                                                                                                 |
| L199-202: the OC values of 8% and 17% corresponding to
the beginning of the second phase and 17% and 27%
corresponding to the non-convergence of the model seem to
not completely fit with figure 5 axes labels. I am wondering
if some mismatch occurred between the axes ticks and
labels.           | There were some problems with the axis in Figure 5 to 7. These have been changed, and the text has been modified accordingly.                                                                                                                                                                                                                                                                                |
| L216-217: this sentence is unclear, please explain better                                                                                                                                                                                                                                                             | These sentences have been modified.                                                                                                                                                                                                                                                                                                                                                                          |
| L225-233: I'm not sure to have correctly understood this point. By looking at Fig. 6, for $\%$ OC = 11% I expect a maximum proportion of the failed contacts of 25% (which corresponds to about 32 contacts). Why is the number of failed contacts between 40 and 70 in figure 8a? I have the same doubt for model 2. | The reviewer is making a really good point.
There was a mistake in the figure and in the text.
It is not 14 contacts but 14% of the joint defined
as OC. This has been modified.                                                                                                                                                                                                                    |
| L234: If 40 models were run, why do figure 8a and 8b report a higher number of points? (at least 70)                                                                                                                                                                                                                  | The scenario 3 has been run 2 times (total of 80 simulations). This has been commented in the paper.                                                                                                                                                                                                                                                                                                         |
| L240: Please comment results also for model 2.                                                                                                                                                                                                                                                                        | Done in the revised manuscript                                                                                                                                                                                                                                                                                                                                                                               |
| L245 "must therefore plays", please correct typo.                                                                                                                                                                                                                                                                     | Done in the revised manuscript                                                                                                                                                                                                                                                                                                                                                                               |
| L255 please correct to "potential unstable blocks"                                                                                                                                                                                                                                                                    | Done in the revised manuscript                                                                                                                                                                                                                                                                                                                                                                               |
| L258 if tensile strength is not considered in the previous analyses, why is it reported in table 3?                                                                                                                                                                                                                   | The sentence L258 is poorly worded. Only
shear failure was observed (local stress reaches
the Mohr-Coulomb failure criterion) while no
tensile failure was reported.                                                                                                                                                                                                                                |
| L265 and table 4: This continuous change of strength
parameters is not scientifically sound, and it should be
supported by a stronger discussion. Please explain why did
Authors not choose a unique set of strength parameters for
all the models                                                        | The difference between model 1 and model 2
has been explained L123: "The failure envelope
properties of RB and RBF (cohesion, friction
angle and tensile strength), were defined to be
close enough to the initial stresses along the
joint". This can be seen in Figure 9 of the initial
version of the paper (Figure 10 in the revised
version of the manuscript). When introducing a |

|                                                                                                                                                                                                                                                                                                                                                                                                                | more realistic tensile strength parameter, we increased the other values to be consistent in our approach.                                                                                                                                                                                                                                                                                                                                                         |
|----------------------------------------------------------------------------------------------------------------------------------------------------------------------------------------------------------------------------------------------------------------------------------------------------------------------------------------------------------------------------------------------------------------|--------------------------------------------------------------------------------------------------------------------------------------------------------------------------------------------------------------------------------------------------------------------------------------------------------------------------------------------------------------------------------------------------------------------------------------------------------------------|
| L269: How do Authors calculate this 6% value? Is it related
to the number of green triangles with normal stress lower
than -10kPa? Please explain                                                                                                                                                                                                                                                        | Exactly, it corresponds to the light green
triangles. It means that 8 contacts present a
tensile normal stress that becomes equal to the
assigned tensile strength.                                                                                                                                                                                                                                                                                       |
| L271 and Figure 9 caption, please correct figure 09 to figure 9                                                                                                                                                                                                                                                                                                                                                | Done in the revised manuscript                                                                                                                                                                                                                                                                                                                                                                                                                                     |
| L277-278: "It justifies that in reality, as the cohesion value
of the rock bridges are 500 times higher than in the study
presented in this paper, only a few portions of rock bridges
allow a potential instable block to be in place". I agree with
this statement; why did not author decide to use more
realistic strength (and rock bridges portion) values? (see
also general comment) | The authors proposed an answer to this relevant
observation. As explained previously, the
objective of this study is the understanding of
the rock bridges failure propagation and the
identification of a cascade-mode behavior. The
choice has been made to consider strength
parameters lower than real ones. This model 3
also comforts us in our approach as the same
behaviour has been observed while considering
higher values. |
| L281: please correct "though" to "through"                                                                                                                                                                                                                                                                                                                                                                     | Done in the revised manuscript                                                                                                                                                                                                                                                                                                                                                                                                                                     |
| L285-287: from a geomechanical point of view it is not
correct to consider failed rock bridges to have the same
strength of intact rock, as well as it is not correct to consider
them to have zero strength. Why did Authors not consider
employing a continuously yielding model?                                                                                                                | This is a really interesting comment from the
reviewer, who suggests considering a material
whose resistance decreases with deformation
(progressive rather than sudden failure). This
would be entirely possible, but it would be a
more complex model and it would require the
introduction of an additional parameter
describing the progressive evolution of
resistance. This could be study in a future study.                        |
| L295-297: Probably it could be interesting to see the propagation of rock bridge failure (like in figure 5) for model 4                                                                                                                                                                                                                                                                                        | A Figure has been added to the paper.                                                                                                                                                                                                                                                                                                                                                                                                                              |
| Figure 2. The gravity direction is in my opinion misleading,
I suggest adding the $40^{\circ}$ slope model. Moreover, it seems to
me that the alpha associations in figure captions are wrong,
i.e. alpha is zero for $80^{\circ}$ (not $40^{\circ}$ ) slope angle and $40^{\circ}$ for
$40^{\circ}$ (not $80^{\circ}$ ) slope angle.                                                              | There was an error in the caption. With this change, the authors feel that the Figure is enough by its own.                                                                                                                                                                                                                                                                                                                                                        |
| Figure 3. Expand the caption to better describe the figure.                                                                                                                                                                                                                                                                                                                                                    | Done in the revised manuscript                                                                                                                                                                                                                                                                                                                                                                                                                                     |
| Figure 5 and figure 7: please put the model 2 sketch on the right side, under the corresponding graph.                                                                                                                                                                                                                                                                                                         | Done in the revised manuscript                                                                                                                                                                                                                                                                                                                                                                                                                                     |

Comments of Reviewer 2

General comments:

Comments on "Cascade effect of rock bridge failure in planar rock slides: explicit numerical modelling with a distinct element code" by Delonca et al. submitted to the journal of NHESS. The authors attempt to estimate the cascade effects of rock bridge failure via DEM. The paper is interesting, but it falls on the borderline regarding the scope of NHESS from my personal point of view. So some more application or relationship regarding to principal/traditional aspect of NHESS should be addressed.

**Response:** The paper proposes to study the phenomenology of rock bridges failure, that has direct influence on the block stability. The author believed that presenting a work related to the rockfall hazard is related to the NHESS journal and of great interest for the readers. This is introduced in the introduction of the manuscript (lines 21 to 33) and discussed in the conclusion part in the last paragraph of the paper.

Specifics comments:

• Why only two aspect /gravity directions from to the same mesh been analyzed. This phenomenon represents to two different sliding behaviors fall vs. slide. But, what about the behavior in transition zone? It should be address to judge the research design of this manuscript/study.

**Response**: The two-behavior analyzed in this paper are shear and tensile failure modes. It is mainly assumed that the first movement preceding the rockfall can be a slide or a topple [14]. While, according to Hutchinson [15] slides can be divided into rotational slides, translational slides and compound slides, all of this types can be, in a first approach, studied as simple sliding. Moreover, intact rock may often fail in tension when considering step-path failure mechanisms [16]. For these reasons, the authors, in a first approach, decided to focus their study on the sliding and shear failure modes. Additionally, to be able to draw clear conclusions about the phenomenon of failure and in particular the cascade effect of the failure, the choice has been made to consider simple cases. More work could be done to support these conclusions, considering for example more complex failure modes, or higher strength parameters.

A paragraph has been added to the conclusion to discuss this point.

- Line 274-275, "the transition phase identified in Figs 5 and 6... and 15%" this sentence not clear identify on the figures or the manuscript.
  - **Response**: The sentence has been rephrased: "For model 3, the transition phase identified previously is comprised between 40% and 50% of the rock joint defined as OC, while in model 1, it is comprised between 10% and 20%".
- The conclusions section: the section should be rewrite to extract the

conclusions, not extract directly from some sentences in each subsection again.

- **Response**: The conclusion section has been rewritten in order to highlight the main conclusions, and not only resume the results of the presented work.
- Fig 5 and 7, typos in Model A and Model B ? Should it be Model 1 and 2?
  - **Response**: Absolutely. The change has been made on both Figures.
- Some limitations should be addressed, e.g. tensile cracks, shear cracks, asperity etc. vs. planner open cracks simplified in this study.
  - **Response**: This is an interesting comment. A paragraph has been added to the conclusion concerning this point.

**Comments of Reviewer 3**

General comments:

Dear Authors I read your paper with great interest. Rock bridges are indeed a "hot topic," and your numerical approach should interest the scientific community. However, I have some comments and some questions that need to be addressed before publication. The other reviewers have already put some forwards, but I have some more.

1) Why is your contact number different from your region's number? What kept you from maintaining them equal? I understand that 100 is a "round number" but that some regions may have more than one contact affects clarity.

**Response**: This choice has been made as a modeling constraint. To be able to randomly select the region using FISH language, it has been necessary to create a table where we could draw random regions using the function *urand*, which allow a random number from uniform distribution between 0.0 and 1.0 to be drawn. Then, when repeating the operation (using a loop), we could select 100 regions (from 0 to 1). Indeed, some regions could have more than one contact, this is why in our results, we present the number of contact and not the number of regions.

2) What random distribution was used to select the regions to transform in OC? Was the relative location of the different regions considered in the randomization? Why did you choose this specific randomization method? How was "10% of the rock joint is intentionally modified from RB to OC contacts"? Is it again through a randomization tool? Of what kind?

**Response**: As explained in the previous answer, a uniform distribution has been considered. The location of the different regions has not been considered in the randomization. This information has been added to the paper.

We choose this specific randomization method has it is the one available in

UDEC (FISH function). When considering 10% of the rock joint, we randomly selected a part of the joint, we count the number of contacts, and we define the proportion. If we consider that we choose the location, in that case we define intentionally the corresponding contacts based on the selected location. This has also been specified in the paper.

3) Is there any account of elastic rebound in your model immediately after a rupture? Response: There is no elastic rebound in our model immediately after a rupture.

4) I also have some issues with the statement in line 211-212 "such a point could be easily be instrumented in the real case of motion tracking" – I have doubts about that, at this scale, since you are modelling displacement of less than a hundredth of a millimeter, and this quantity is really difficult to monitor on field. Moreover, what do you mean by "Figure 7 shows that there is only one trend when considering displacements" (line 216)?

**Response**: This is a really interesting comment from the reviewer. The idea was to select a point that could be easily instrumented if displacements of the order of mm were observed in our simulation before the failure of the block. In our simulation it is not the case, which shows that tracking displacements may not be the best way of assessing the rock bridges failure in the field. A comment has been added in lines 211-213 and in the conclusion.

Regarding Figure 7, a line has been added to highlight the commented "trend".

5) Finally, the limitations of 2D approaches to study rock bridges should be discussed. We know that in 2D rock bridges are scale dependent (Elmo, 2018). Do you believe that the reduction of the strength parameters you imposed was scaled accordingly with your geometry? How would a tri-dimensional geometry affect your simulations?

**Response**: Elmo et al. [16] showed the following limitation in 2D models to characterize rock bridges: The definition of rock bridge length given in the literature for the 2D case is scale dependent and controlled by the height of the slope and the dip of the potential failure surface. Accordingly, equivalent strength parameters should be modified to account for scale effects and the reduction of intact rock strength with increasing rock bridge length. The authors proposed equations to evaluate equivalents cohesion and friction angle of the equivalent discontinuity which considers both the rock bridges and the joint surface characteristics. Considering equivalent discontinuity parameters allow the scale effect to be integrated in the analysis. While this is a really interesting approach, it is not quite applicable here as we have really low percentage of rock bridges (K parameter close to 1). Based on the work of Elmo et al., it seems that we underestimate the strength parameter reduction, and it was not scaled accordingly with our geometry (we discussed previously the choice of our parameters).

Moreover, in our study, we did not try to reproduce a 3D case, but we focused

only on a 2D case. Regarding a 3D approach, the author think that the observed phenomenology should be the same, but it would allow us to vary on a better way the position and proportion of rock bridges. However, the authors do not think that the results of the paper would be different, but another study would be required to confirm or inform this.

Specifics comments:

- The scheme in fig 1a is not clear and readable; please emend it maybe using colors Why is g horizontal in figure 2? Please show the mesh of your model.
  - **Response**: Both Figure 1 and Figure 2 has been modified based on the comment. g is not horizontal in the Figure.
- Fig 3 caption must be way more informative. The graphs on the left should be described, especially the Step 2 one where we see two evolutions (?) one in black and one in gray
  - **Response**: The caption has been modified.
- The caption of Fig 4 should include the fact that it refers to Scenario 1, as Scenario 2 in the caption of Fig.2
  - **Response**: Information relative to the scenarios has been added to each caption (from Fig4 to Fig 9).
- In Fig. 5 and fig 6 maybe substitute "the block does not stabilize" with "collapse" or "non-convergence of the simulation"
  - **Response**: In all figures "the block does not stabilize" has been changed by "non-convergence of the simulation".
- Fig. 8. Wouldn't it be better to plot the height (better than altitude) of OC using the unstable block's base as reference? So, a length that spans between 0 and 6\*cos(alpha)? In this way, it would be easier to compare the average height of the contacts with the block's median point.

Why do you think in model 2 the average height of the OC that induce collapse is higher (7 m) than the median point of the block?

And why they seem to coincide in model 1? This needs to be discussed. It would also be useful if you plot a histogram of the average heights. Finally, the unit of measures [m] should be indicated in the figure.

• **Response**: The Figure has been modified as proposed by the reviewer.

Regarding the model 2, the presented line represents the average heigh of the OC (it may be confusing, the figure has been clarified). Same for model 1. In both cases, we are presenting the number of failed contacts for a same number of contacts defined as open crack for the different simulation (18 contacts for model 1 and 46 contacts for model 2). It shows that the average location of these open crack has a direct influence on the number of rock bridges that failed due to increase of shear stresses along the joint.

A histogram has been added and discussed in the revised version of the paper and enclosed to this answer.

Fig. 9. Histogram of the average altitude ("y" coordinate) of the OC contacts, for models 1 and 2 considering Scenario 3 (runs 2 times). N is equal to 18 and 46 respectively for model 1 and 2.

- Fig 9 (not 09) Scenario 3, not model 3
   Response: The change has been made.
- In general, for many figures, the unit of measurement should be indicated in [] brackets instead of "in kPa"
  - **Response**: This has been modified in the Figures.
- L 23 in other words
  - o **Response**: Done in the revised manuscript
- L 33 are rock bridges
  - **Response**: Done in the revised manuscript
- L 100 "This division has been undertaken using FISH"
   Response: Done in the revised manuscript
- L 245 play
  - **Response**: Done in the revised manuscript

- L 249 Figure 8 (no "the")
  - **Response**: Done in the revised manuscript
- Check reference format: Brideau, Delonca (journal name abbreviated), Spreafico
  - **Response**: The entire manuscript has been revised regarding the references

**Bibliography**

[1] M. Frayssines and D. Hantz, "Modelling and back-analysing failures in steep limestone cliffs," *Int. J. Rock Mech. Min. Sci.*, vol. 46, no. 7, pp. 1115–1123, 2009.
[2] B. Matasci, M. Jaboyedoff, L. Ravanel, and P. Deline, "Stability Assessment, Potential Collapses and Future Evolution of the West Face of the Drus (3,754 m a.s.l., Mont Blanc Massif)," in *Engineering Geology for Society and Territory - Volume 2*, Cham, 2015, pp. 791–795, doi: 10.1007/978-3-319-09057-3\_134.

[3] Z. Tuckey and D. Stead, "Improvements to field and remote sensing methods for mapping discontinuity persistence and intact rock bridges in rock slopes," *Eng. Geol.*, vol. 208, pp. 136–153, Jun. 2016, doi: 10.1016/j.enggeo.2016.05.001.

[4] J. Kemeny, "Time-dependent drift degradation due to the progressive failure of rock bridges along discontinuities," *Int. J. Rock Mech. Min. Sci.*, vol. 42, no. 1, pp. 35–46, Jan. 2005, doi: 10.1016/j.ijrmms.2004.07.001.

[5] E. Eberhardt, D. Stead, and J. S. Coggan, "Numerical analysis of initiation and progressive failure in natural rock slopes—the 1991 Randa rockslide," *Int. J. Rock Mech. Min. Sci.*, vol. 41, no. 1, pp. 69–87, Jan. 2004, doi: 10.1016/S1365-1609(03)00076-5.

[6] C. Scavia, "A method for the study of crack propagation in rock structures," *Géotechnique*, vol. 45, no. 3, pp. 447–463, Sep. 1995, doi: 10.1680/geot.1995.45.3.447.
[7] M.-A. Brideau, M. Yan, and D. Stead, "The role of tectonic damage and brittle rock fracture in the development of large rock slope failures," *Geomorphology*, vol. 103, no. 1, pp. 30–49, Jan. 2009, doi: 10.1016/j.geomorph.2008.04.010.

[8] V. Bonilla–Sierra, L. Scholtès, F. Donzé, and M. Elmouttie, "DEM analysis of rock bridges and the contribution to rock slope stability in the case of translational sliding failures," *Int. J. Rock Mech. Min. Sci.*, vol. 80, pp. 67–78, Dec. 2015, doi: 10.1016/j.ijrmms.2015.09.008.

[9] B. Harthong, L. Scholtès, and F.-V. Donzé, "Strength characterization of rock masses, using a coupled DEM–DFN model," *Geophys. J. Int.*, vol. 191, no. 2, pp. 467–480, Nov. 2012, doi: 10.1111/j.1365-246X.2012.05642.x.

[10] G. G. D. Zhou, P. Cui, X. Zhu, J. Tang, H. Chen, and Q. Sun, "A preliminary study of the failure mechanisms of cascading landslide dams," *Int. J. Sediment Res.*, vol. 30, no. 3, pp. 223–234, Sep. 2015, doi: 10.1016/j.ijsrc.2014.09.003.

[11] G. Pappalardo and S. Mineo, "Rockfall hazard and risk assessment: the promontory of the pre-Hellenic village Castelmola case, north-eastern Sicily (Italy)," in *Engineering Geology for Society and Territory-Volume 2*, Springer, 2015, pp. 1989–1993.

[12] S. Mineo, G. Pappalardo, M. Mangiameli, S. Campolo, and G. Mussumeci, "Rockfall analysis for preliminary hazard assessment of the cliff of Taormina Saracen Castle (Sicily)," *Sustainability*, vol. 10, no. 2, p. 417, 2018.

[13] R. A. Kromer, E. Rowe, J. Hutchinson, M. Lato, and A. Abellán, "Rockfall risk

management using a pre-failure deformation database," *Landslides*, vol. 15, no. 5, pp. 847–858, 2018.

[14] D. M. Cruden and D. J. Varnes, "Landslides: Investigation and Mitigation. Chapter 3-Landslide types and processes," *Transp. Res. Board Spec. Rep.*, no. 247, 1996.

[15] J. N. Hutchinson, "General report: morphological and geotechnical parameters of landslides in relation to geology and hydrogeology," in *International Journal of Rock Mechanics and Mining Sciences & Geomechanics Abstracts*, Lausanne, Jul. 1988, vol. 26, p. 88.

[16] D. Elmo, D. Donati, and D. Stead, "Challenges in the characterisation of intact rock bridges in rock slopes," *Eng. Geol.*, vol. 245, 2018, doi: 10.1016/j.enggeo.2018.06.014.

---

## Author Response (AR2)

Santiago, Chile, March 2021

Authors' Response

*Nhess- 2020-279 - Cascade effect of rock bridge failure in planar rock slides: explicit numerical modelling with a distinct element code.*

Dear Editor,

The authors thank the 2 reviewers that took time to review the revised version of our manuscript. Referee #1 does not have further comments. Referee #3 added some interesting comments in relation with the use of the ITASCA code. In the following we present a response to its comments. Moreover, we will submit a revised version of the manuscript, with the modifications highlighted in the red color.

Comments of Referee #3

Dear authors, your answers for points 1 and 2 make sense but show a not so great familiarity with the ITASCA codes. In fact, all the issues regarding regions and stochastic distribution can be easily bypassed by preparing a simple script in Matlab (or whichever scripting code you like) to generate the stochastic configurations using every random distribution you prefer (Bossi et al, 2016, Engineering Geology) and also to modify the geometry. This approach would have significantly improved your paper since it would have been possible to account for some contiguity effect using different distributions with respect to an uniform one. And should be certainly considered for future works.

> **Response**: This is a really good point. We added a paragraph in the conclusion of the second revised version of our paper to highlight this limitation. It will certainly be taking into account for further work.

Thank you for all the other answers and for incorporating the considerations about monitoring in the new version. In the light of some limitations of your work, that are now honestly stated in the text, I would advise you to change the title to "Cascade effect of rock bridge failure in planar rock slides: numerical test with a distinct element code".

> **Response**: Thank you for this proposal of title. We have changed the title accordingly.